

# Simulation of the chemical evolution of biomass burning organic aerosol

Georgia N. Theodoritsi[1,2] and Spyros N. Pandis[1,2,3]

[1]*Department of Chemical Engineering, University of Patras, Patras, Greece*

[2]*Institute of Chemical Engineering Sciences, Foundation for Research and Technology Hellas (FORTH/ICE-HT), Patras, Greece*

[3]*Department of Chemical Engineering, Carnegie Mellon University, Pittsburgh, PA 15213, USA*

**Abstract**

The chemical transport model PMCAMx was extended to investigate the effects of partitioning and photochemical aging of biomass burning emissions on organic aerosol (OA) concentrations. A source-resolved version of the model, PMCAMx-SR, was developed in which biomass burning organic aerosol (bbOA) and its oxidation products are represented separately from the other OA components. The volatility distribution of bbOA and its chemical aging were simulated based on recent laboratory measurements. PMCAMx-SR was applied to Europe during an early summer (1-29 May 2008) and a winter period (25 February-22 March 2009).

During the early summer, the contribution of biomass burning (both primary and secondary species) to total OA levels over continental Europe was estimated to be approximately 16%. During winter the same contribution was nearly 47% due to both extensive residential wood combustion, but also wildfires in Portugal and Spain. The intermediate volatility compounds (IVOCs) with effective saturation concentration values of $10^5$ and $10^6$ µg m$^{-3}$ are predicted to contribute around one third of the bbOA during the summer and 15% during the winter by forming secondary OA. The uncertain emissions of these compounds and their SOA formation potential require additional attention. Evaluation of PMCAMx-SR predictions against aerosol mass spectrometer measurements in several sites around Europe suggests reasonably good performance for OA (fractional bias less than 35% and fractional error less than 50%). The performance was weaker during the winter suggesting uncertainties in the residential heating emissions and the simulation of the resulting bbOA in this season.



## 1   Introduction


Atmospheric aerosols, also known as particulate matter (PM), are suspensions of
fine solid or liquid particles in air. These particles range in diameter from a few
nanometers to tens of micrometers. Atmospheric particles contain a variety of non-
volatile and semi-volatile compounds including water, sulfates, nitrates, ammonium,
dust, trace metals, and organic matter. Many studies have linked increased mortality
(Dockery et al., 1993), decreased lung function (Gauderman et al., 2000), bronchitis
incidents (Dockery et al., 1996), and respiratory diseases (Pope, 1991; Schwartz et al.,
1996; Wang et al., 2008) with elevated PM concentrations. The most readily
perceived impact of high particulate matter concentrations is visibility reduction in
polluted areas (Seinfeld and Pandis, 2006). Aerosols also play an important role in the
energy balance of our planet by scattering and absorbing radiation (Schwartz et al.,

45   1996).

Organic aerosol (OA) is a major component of fine PM in most locations
around the world. More than 50% of the atmospheric fine aerosol mass is comprised
of organic compounds at continental mid-latitudes and as high as 90% in tropical
forested areas (Andreae and Crutzen, 1997; Roberts et al., 2001; Kanakidou et al.,
2005). Despite their importance, there are many remaining questions regarding their
identity, chemistry, lifetime, and in general fate of these organic compounds. OA
originates from many different anthropogenic and biogenic sources and processes and
has been traditionally categorized into primary OA (POA) which is directly emitted
into the atmosphere as particles and secondary OA (SOA) that is formed from the
condensation of the oxidation products of volatile (VOCs), intermediate volatility
(IVOCs), and semivolatile organic compounds (SVOCs). Both POA and SOA are
usually characterized as anthropogenic (aPOA, aSOA) and biogenic (bPOA, bSOA)
depending on their sources. In this work we define biomass burning OA (bbOA) as
the sum of bbPOA and bbSOA following the terminology proposed by Murphy et al.

60   (2014).

Biomass burning is an important global source of air pollutants that affect
atmospheric chemistry, climate, and environmental air quality. In this work, the term
biomass burning includes wildfires, prescribed burning in forests and other areas,
residential wood combustion for heating and other purposes, and agricultural waste
burning. Biomass burning is a major source of particulate matter, nitrogen oxides,
carbon monoxide, volatile organic compounds, as well as other hazardous air



pollutants. Biomass burning contributes around 75% of global combustion POA
(Bond et al., 2004). In Europe, biomass combustion is one of the major sources of
OA, especially during winter (Puxbaum et al., 2007; Gelencser et al., 2007).
Chemical transport models (CTMs) have traditionally treated POA emissions
as non-reactive and non-volatile. However, dilution sampler measurements have
indicated that POA is clearly semi-volatile (Lipsky and Robinson, 2006; Robinson et
al., 2007; Huffman et al., 2009a, 2009b). The semi-volatile character of POA
emissions can be described by the volatility basis set (VBS) framework (Donahue et
al., 2006; Stanier et al., 2008). The VBS is a scheme of simulating OA accounting for
changes in gas-particle partitioning due to dilution, temperature changes, and
photochemical aging. The third Fire Lab at Missoula Experiment (FLAME-III)
investigated a suite of fuels associated with prescribed burning and wildfires (May et
al., 2013). The bbOA partitioning parameters derived from that study are used in this
work to simulate the dynamic gas-particle partitioning and photochemical aging of
bbOA emissions.
A number of modeling efforts have examined the contribution of the semi-
volatile bbOA emissions to ambient particulate levels using the VBS framework. For
example, Fountoukis et al. (2014) used a three dimensional CTM with an updated
wood combustion emission inventory distributing OA emissions using the volatility
distribution proposed by Shrivastava et al. (2008). However, this study assumed the
same volatility distribution for all OA sources. This volatility distribution is not in
general representative of biomass burning emissions since it was derived based on
experiments using fossil fuel sources (Shrivastava et al., 2008).
The main objective of this study is to develop and test a CTM treating biomass
burning organic aerosol (bbOA) emissions separately from all the other anthropogenic
and biogenic emissions. This extended model should allow at least in principle more
accurate simulation of OA and direct predictions of the role of bbOA in regional air
quality. The rest of the manuscript is organized as follows. First, a brief description of
the new version of PMCAMx (PMCAMx-SR) is provided. The source-resolved
version of PMCAMx (PMCAMx-SR) treats bbOA emissions and their chemical
reactions separately from those of other OA sources. The details of the application of
PMCAMx-SR in the European domain for a summer and a winter period are
presented. In the next section, the predictions of PMCAMx-SR are evaluated using


AMS measurements collected in Europe. Finally, the sensitivity of the model to
different parameters is quantified.

## 2   PMCAMx-SR description

PMCAMx-SR is a source-resolved version of PMCAMx (Murphy and Pandis,
2009; Tsimpidi et al., 2010; Karydis et al., 2010), a three-dimensional chemical
transport model that uses the framework of CAMx (Environ, 2003) and simulates the
processes of horizontal and vertical advection, horizontal and vertical dispersion, wet
and dry deposition, gas, aqueous and aerosol-phase chemistry. The chemical
mechanism employed to describe the gas-phase chemistry is based on the SAPRC
mechanism (Carter, 2000; Environ, 2003). The version of SAPRC currently used
includes 211 reactions of 56 gases and 18 radicals. The SAPRC mechanism has been
updated to include gas-phase oxidation of semivolatile organic compounds (SVOCs),
intermediate volatility organic compounds (IVOCs). Three detailed aerosol modules
are used to simulate aerosol processes: inorganic aerosol growth (Gaydos et al., 2003;
Koo et al., 2003), aqueous phase chemistry (Fahey and Pandis, 2001), and secondary
organic aerosol (SOA) formation and growth (Koo et al., 2003). The above modules
use a sectional approach to dynamically track the size evolution of each aerosol
component across 10 size sections spanning the diameter range from 40 nm to 40 μm.

### 2.1   Organic aerosol modelling

PMCAMx-SR simulates organic aerosol based on the volatility basis set (VBS)
framework (Donahue et al., 2006; Stanier et al., 2008). VBS is a unified scheme of
treating organic aerosol, simulating the volatility, gas-particle partitioning, and
photochemical aging of organic pollutant emissions. PMCAMx-SR incorporates
separate VBS variables and parameters for the various OA components based on their
source.

### 2.1.1   Volatility of primary emissions

PMCAMx-SR assumes that all primary emissions are semi-volatile.
According to the VBS scheme, species with similar volatility are lumped into bins
expressed in terms of effective saturation concentration values, $C^*$, separated by
factors of 10 at 298 K. POA emissions are distributed across a nine-bin VBS with $C^*$
values ranging from $10^{-2}$ to $10^6$ μg m$^{-3}$ at 298 K. SVOCs and IVOCs are distributed





among the 1, 10, 100 µg m$^{-3}$ $C^*$ bins and 10$^3$, 10$^4$, 10$^5$, 10$^6$ µg m$^{-3}$ $C^*$ bins
respectively. Table 1 lists the generic POA volatility distribution proposed by
Shrivastava et al. (2008) assuming that the IVOC emissions are approximately equal
to 1.5 times the primary organic aerosol emissions (Robinson et al., 2007; Tsimpidi et
al., 2010; Shrivastava et al., 2008). This volatility distribution is used in PMCAMx-
SR for all sources with the exception of wood burning. In the original PMCAMx this
volatility distribution is also used for wood burning emissions.
The partitioning calculations of primary emissions are performed using the
same module used to calculate the partitioning of all semi-volatile organic species
(Koo et al., 2003). This is based on absorptive partitioning theory and assumes that
the bulk gas and particle phases are in equilibrium and that all condensable organics
form a pseudo-ideal solution (Odum et al., 1996; Strader et al., 1999). Organic gas-
particle partitioning is assumed to depend on temperature and aerosol composition.
The Clausius-Clapeyron equation is used to describe the effects of temperature on $C^*$
and partitioning. Table 1 also lists the enthalpies of vaporization currently used in
PMCAMx and PMCAMx-SR. All POA species are assumed to have an average
molecular weight of 250 g mol$^{-1}$.

### 2.1.2    Secondary organic aerosol from VOCs

Following Lane et al. (2008a), the SOA VBS-scheme uses four surrogate SOA
compounds for each VOC precursor with 4 volatility bins (1, 10, 100, 1000 µg m$^{-3}$) at
298 K. Anthropogenic (aSOA-v) and biogenic (bSOA-v) components are simulated
separately. aSOA components are assumed to have an average molecular weight of
150 g mol$^{-1}$, while bSOA species 180 g mol$^{-1}$. Laboratory results from the smog-
chamber experiments of Ng et al. (2006) and Hildebrandt et al. (2009) are used for the
anthropogenic aerosol yields.

### 2.1.3    Chemical aging mechanism

All OA components are treated as chemically reactive in PMCAMx-SR.
Vapors resulting from the evaporation of POA are assumed to react with OH radicals
with a rate constant of $k = 4 \times 10^{-11}$ cm$^3$ molec$^{-1}$ s$^{-1}$ resulting in the formation of
oxidized OA. These reactions are assumed to lead to an effective reduction of
volatility by one order of magnitude. Semi-volatile SOA components are also
assumed to react with OH in the gas phase with a rate constant of $k = 1 \times 10^{-11}$ cm$^3$



molec$^{-1}$ s$^{-1}$ for anthropogenic SOA (Atkinson and Arey, 2003). Biogenic SOA aging
is assumed to lead to zero net change of volatility (Lane et al., 2008b). Each reaction
is assumed to increase the OA mass by 7.5% to account for added oxygen.

**2.2   PMCAMx-SR enhancements**
In PMCAMx-SR, the fresh biomass burning organic aerosol (bbOA) and its
secondary oxidation products (bbSOA) are simulated separately from the other POA
components. The May et al. (2013) volatility distribution is used to simulate the gas-
particle partitioning of fresh bbOA. This distribution includes surrogate compounds
up to a volatility of $10^4$ μg m$^{-3}$. This means that the more volatile IVOCs, which could
contribute to SOA formation, are not included. To close this gap, the values of the
volatility distribution of Robinson et al. (2007) are used for the $10^5$ and $10^6$ μg m$^{-3}$
bins (Table 1). The sensitivity of PMCAMx-SR to the IVOC emissions added to the
May et al. (2013) distribution will be explored in a subsequent section. The effective
saturation concentrations and the enthalpies of vaporization used for bbOA in
PMCAMx-SR are also listed in Table 1. The new bbOA scheme requires the
introduction of 36 new organic species to simulate both phases of fresh primary and
oxidized bbOA components. The rate constant used for the chemical aging reactions
is the same as the one currently used for all POA components and has a value of $k = 4$
$\times$ $10^{-11}$ cm$^3$ molec$^{-1}$ s$^{-1}$. The volatility distributions of bbOA in PMCAMx and
PMCAMx-SR are shown in Fig. 1a. The volatility distribution implemented in
PMCAMx-SR results in less volatile bbOA for ambient OA levels (a few μg m$^{-3}$)
(Fig. 1b).

**3   Model application**
PMCAMx-SR was applied to a 5400×5832 km$^2$ region covering Europe with
36×36 km grid resolution and 14 vertical layers extending up to 6 km. The model was
set to perform simulations on a rotated polar stereographic map projection. The
necessary inputs to the model include horizontal wind components, temperature,
pressure, water vapor, vertical diffusivity, clouds, and rainfall. All meteorological
inputs were created using the meteorological model WRF (Weather Research and
Forecasting) (Skamarock et al., 2005). The simulations were performed during a



summer (1-29 May 2008) and a winter period (25 February-22 March 2009). In order to limit the effect of the initial conditions on the results, the first two days of each simulation were excluded from the analysis.

Anthropogenic and biogenic emissions in the form of hourly gridded fields were developed both for gases and primary particulate matter. Anthropogenic gas emissions include land emissions from the GEMS dataset (Visschedijk et al., 2007) and also emissions from international shipping activities. Anthropogenic particulate matter mass emissions of organic and elemental carbon are based on the Pan-European Carbonaceous Aerosol Inventory (Denier van der Gon et al., 2010) that has been developed as part of the EUCAARI project activities (Kulmala et al., 2009). All relevant significant emission sources are included in the two inventories. Emissions from ecosystems were calculated offline by MEGAN (Model of Emissions of Gases and Aerosols from Nature) (Guenther et al., 2006). The marine aerosol emission model developed by O'Dowd et al. (2008) has been used to estimate mass fluxes for both accumulation and coarse mode including the organic aerosol fraction. Wind speed data from WRF and chlorophyll-a concentrations are the inputs needed for the marine aerosol emissions module.

Day-specific wildfire emissions were also included (Sofiev et al., 2008a; 2008b). Anthropogenic sources of wood combustion include residential heating and agricultural activities. The gridded emission inventories of bbOA species for the two modeled periods are shown in Fig. 2. During the early summer simulated period wildfires were responsible for 60% of the bbOA emissions, agricultural waste burning for 15% and residential wood combustion for 25% (Table 2). Details about the OA emission rates from agricultural activities are provided in the Supplementary Information (Fig. S1). During winter residential combustion is the dominant source (63%). The wintertime wildfire emissions in the inventory, approximately 3,000 tn $d^{-1}$, are quite high especially when compared with the corresponding summer value which is 1,700 tn $d^{-1}$. The spatial distribution of OA emission rates from wildfires during 25 February-22 March 2009 is provided in the Supplementary Information (Fig. S2). Analysis of fire counts in satellite observations used for the development of the inventory suggests that some agricultural emissions have probably been attributed to wildfires. All bbOA sources are treated the same way in PMCAMx-SR so this potential misattribution does not affect our results.



## 4    PMCAMx-SR testing

To test our implementation of the source-resolved VBS in PMCAMx-SR we compared its results with those of PMCAMx using the same VBS parameters. For this test we used in PMCAMx-SR the default PMCAMx bbOA partitioning parameters shown in Table 1 as proposed by Shrivastava et al. (2008). In this way both models should simulate the bbOA in exactly the same way, but PMCAMx-SR describes it independently while PMCAMx lumps it with other primary OA. The differences between the corresponding OA concentrations predicted by the two models were on average less than $10^{-3}$ μg m$^{-3}$ (0.03%). The maximum difference was approximately 0.03 μg m$^{-3}$ (0.6%) in western Germany. This suggests that our changes to the code of PMCAMx to develop PMCAMx-SR did not introduce any inconsistencies with the original model. The small differences are due to numerical issues in the advection/dispersion calculations.

## 5    Contribution of bbOA to PM over Europe

In this section the predictions of PMCAMx-SR for the base case simulations during 1 - 29 May 2008 and 25 February - 22 March 2009 are analysed. Figure 3 shows the PMCAMx-SR predicted average ground-level PM$_{2.5}$ concentrations for the various OA components for the two simulated periods.

The POA from non-bbOA sources will be called fossil POA (fPOA) in the rest of the paper. fPOA levels over Europe were on average around 0.1 μg m$^{-3}$ during both periods (Figs. 3a and 3b). However, their spatial distributions are quite different. During May, predicted fPOA concentrations are as high as 2 μg m$^{-3}$ in polluted areas in central and northern Europe but are less than 0.5 μg m$^{-3}$ in the rest of the domain. These low levels are due to the evaporation of POA in this warm period. For the winter period peak fPOA levels are higher reaching values of around 3.5 μg m$^{-3}$ in Paris and Moscow. fPOA contributes approximately 3.5% and 6% to total OA in Europe during May 2008 and February-March 2009 respectively. bbPOA concentrations have peak average values 7 μg m$^{-3}$ in St. Petersburg in Russia and 10 μg m$^{-3}$ in Porto in Portugal during summer and winter respectively (Figures 3c and 3d). During the summer bbPOA is predicted to contribute 5% to total OA, and its contribution during winter increases to 32%. The average predicted bbOA concentrations over Europe are 0.1 μg m$^{-3}$ and 0.8 μg m$^{-3}$ during the summer and the winter period respectively.



The SOA resulting from the oxidation of IVOCs (SOA-iv) and evaporated
POA (SOA-sv) has concentrations as high as 1 μg m$^{-3}$ in central Europe and the
average levels are around 0.3 μg m$^{-3}$ (13% contribution to total OA) during summer
(Fig. 3e). During winter the peak concentration value was a little less than 0.5 μg m$^{-3}$
in Moscow in Russia and the average levels were approximately 0.1 μg m$^{-3}$ (5.5%
contribution to total OA) (Fig 3f). The highest average concentration of bbSOA-sv
and bbSOA-iv (biomass burning SOA from intermediate volatility and semi-volatile
precursors) was approximately 1 μg m$^{-3}$ in Lecce in Italy during summer and 3.5 μg
m$^{-3}$ in Porto during winter. During May bbSOA is predicted to contribute 11% to total
OA over Europe and during February-March 2009 its predicted contribution is 15%.
The average bbSOA is 0.3 μg m$^{-3}$ during summer and approximately 0.4 μg m$^{-3}$
during winter (Figs. 3g and 3h). During the summer, the remaining 67% of total OA is
biogenic SOA (52%) and anthropogenic SOA (15%), and in winter of the remaining
41% of total OA, 36% is biogenic and 5% is anthropogenic SOA (not shown).
In areas like St. Petersburg in Russia predicted hourly bbOA levels exceeded
300 μg m$^{-3}$ due to the nearby fires affecting the site on May 3-5 (Fig. 4). For these
extremely high concentrations most of the bbOA (90% for St. Petersburg) was
primary with the bbSOA contributing around 10%. The spatiotemporal evolution of
bbPOA and bbSOA during May 1–6 in Scandinavia and northwest Russia is depicted
in Figure 5. A series of fires started in Russia on May 1, becoming more intense
during the next days until May 6 when they were mostly extinguished. bbSOA, as
expected, follows the opposite evolution with low concentration values in the
beginning of the fire events (May 1) and higher values later on. The bbSOA
production increases the range of influence of the fires.
In Majden (FYROM) fires contributed up to 25 μg m$^{-3}$ of bbOA on May 25-
26. The bbSOA was 15% of the bbOA in this case (Fig. S3). Fires also occurred in
south Italy (Catania) and contributed up to 52 μg m$^{-3}$ of OA on May 15-17. During
this period the bbSOA was 13% of the bbOA (Fig. S3). Paris (France) and Dusseldorf
(Germany) were further away from major fires but were also affected by fire
emissions during most of the month (Fig. S3). The maximum hourly bbOA levels in
these cities were around 5 μg m$^{-3}$, but bbSOA in this case represents according to the
model around 35% of the total bbOA in Paris and 55% in Dusseldorf.
During the winter simulation period, there were major fires during March 20-
22 in Portugal and northwestern Spain. The maximum predicted hourly bbOA





concentration in Porto (Portugal) exceeded 700 µg m$^{-3}$ on March 21. During the same
3 days in March the average levels of bbPOA in Portugal and Spain were 9 µg m$^{-3}$
and their contribution to total OA was 62%. bbPOA was 80% of the total bbOA
during March 20-22 in the Iberian Peninsula.

## 6   Role of the more volatile IVOCs

We performed an additional sensitivity simulation where we assumed that there

are no emissions of more volatile IVOCs (those in the $10^5$ and $10^6$ µg m$^{-3}$ bins). The
partitioning parameters used in this sensitivity test are shown in Table 1. The
emissions rates for each volatility bin during the two modeled periods are provided in
the Supplementary Information (Table S1). The absolute emissions assigned to the
lower volatility bins are approximately the same for both simulations. More
specifically, during May 2008, the emission rates of LVOCs ($10^{-2}$, $10^{-1}$ µg m$^{-3}$ $C^*$
bins) and SVOCs ($10^0$, $10^1$, $10^2$ µg m$^{-3}$ $C^*$ bins) are 530 and 1050 tn d$^{-1}$ respectively
for the base-case run and 580 and 1160 tn d$^{-1}$ respectively for the sensitivity run.
During February-March 2009, the emission rates of LVOCs and SVOCs are 2100 and
4100 tn d$^{-1}$ respectively for the base-case run and 2300 and 4500 tn d$^{-1}$ respectively
for the sensitivity run. The base case simulation assumes higher emissions in the
upper volatility bins of the IVOCs ($10^3$, $10^4$, $10^5$, $10^6$ µg m$^{-3}$ $C^*$ bins) which can be
converted to bbSOA. During summer, the emission rate of IVOCs is 4460 tn d$^{-1}$ in the
base-case run and 1160 tn d$^{-1}$ in the sensitivity test. During winter, the emission rate
of IVOCs is 17400 tn d$^{-1}$ in the base case and 4500 tn d$^{-1}$ in the sensitivity test.

The base case and the sensitivity simulations predict practically the same

bbPOA concentrations in both periods (Fig. 6) as expected based on the emission
inventory. During summer, the average absolute change of bbPOA in Europe is
around 10% (corresponding to 0.01 µg m$^{-3}$) (Fig. 6a). The average difference in
bbSOA is significantly higher and around 60% (0.2 µg m$^{-3}$ on average) due to the
higher IVOC emissions of the base case simulation. The atmospheric conditions
during this warm summer period (high temperature, UV radiation, relative humidity)
lead to high OH concentrations and rapid production of bbSOA.

During winter, the average absolute change for both bbPOA and bbSOA in

Europe is approximately 0.1 µg m$^{-3}$ (Fig. 6b and 6f). These correspond to 15% change
for the primary and 25% for the secondary bbOA levels. The maximum difference for
average bbPOA is approximately 5 µg m$^{-3}$ and for bbSOA around 1.5 µg m$^{-3}$ both in





northwestern Portugal. However, during the fire period (March 20-22) in Spain and
Portugal the maximum concentration difference between the two cases was 20 µg m$^{-3}$
for bbPOA and 7 µg m$^{-3}$ for bbSOA.

Figure 7 shows the total bbOA (sum of bbPOA and bbSOA) during both

periods. Higher bbOA concentrations are predicted in the base case simulation due to
the higher bbSOA concentrations from higher IVOC emissions. During summer the
contributions of the biomass burning IVOC oxidation products to total bbOA exceed
30% over most of Europe, while during winter these components are important
mostly over Southern Europe and the Mediterranean (Fig. S4).

**7     Comparison with field measurements**

In order to assess the PMCAMx-SR performance during the two simulation

periods the model's predictions were compared with AMS hourly measurements that
took place in several sites around Europe. All observation sites are representative of
regional atmospheric conditions.

The PMF technique (Paatero and Tapper, 1994; Lanz et al., 2007; Ulbrich et

al., 2009; Ng et al., 2010) was used to analyze the AMS organic spectra providing
information about the sources contributing to the OA levels (Hildebrandt et al., 2010;
Morgan et al., 2010). The method classifies OA into different types based on different
temporal emission and formation patterns and separates it into hydrocarbon-like
organic aerosol (HOA, a POA surrogate), oxidized organic aerosol (OOA, a SOA
surrogate) and fresh bbOA. Additionally, factor analysis can further classify OOA
into more and less oxygenated OOA components. Fresh bbOA can be compared
directly to the PMCAMx-SR bbPOA predictions, whereas bbSOA should, in principle
at least, be included in the OOA factors. The AMS HOA can be compared with
predicted fresh POA. The oxygenated AMS OA component can be compared against
the sum of anthropogenic and biogenic SOA (aSOA, bSOA), SOA-sv and SOA-iv,
bbSOA and OA from long range transport.

PMCAMx-SR performance is quantified by calculating the mean bias (MB),

the mean absolute gross error (MAGE), the fractional bias (FBIAS), and the fractional
error (FERROR) defined as:

$$MB = \frac{1}{n}\sum_{i=1}^{n}(P_i - O_i) \qquad\qquad MAGE = \frac{1}{n}\sum_{i=1}^{n}|P_i - O_i|$$



$$\text{FBIAS} = \frac{2}{n}\sum_{i=1}^{n}\frac{P_i - O_i}{P_i + O_i} \qquad \text{FERROR} = \frac{2}{n}\sum_{i=1}^{n}\frac{\left|P_i - O_i\right|}{P_i + O_i}$$


where $P_i$ is the predicted value of the pollutant concentration, $O_i$ is the observed value
and $n$ is the number of measurements used for the comparison. AMS measurements
are available in 4 stations (Cabauw, Finokalia, Melpitz and Mace Head) during 1-29
May 2008 and 7 stations (Cabauw, Helsinki, Mace Head, Melpitz, Hyytiala, Barcelona and
Chilbolton) during 25 February-23 March 2009.

During May 2008 a bbPOA factor was identified based on the PMF analysis

of the measurements only in Cabauw and Mace Head. In the other two sites
(Finokalia and Melpitz) PMCAMx-SR predicted very low average bbPOA levels (less
than 0.1 μg m$^{-3}$), so its predictions for these sites can be viewed as consistent with the
results of the PMF analysis. Figure 8 shows the comparison of the predicted bbPOA
by PMCAMx-SR with the observed values in Cabauw. The average AMS-PMF bbOA
was 0.4 μg m$^{-3}$ and the predicted average bbPOA by PMCAMx-SR was also 0.4 μg
m$^{-3}$. The mean bias was only -0.01 μg m$^{-3}$. The model however tended to overpredict
during the first 10 days and to underpredict during the last week. In Mace Head
PMCAMx-SR predicts high bbOA levels during May 14 – 15, but unfortunately the
available measurements started on May 16. During the last two weeks of the
simulation the model predicts much lower bbOA levels (approximately 0.35 μg m$^{-3}$
less) than the AMS-PMF analysis. The same problem was observed in Cabauw
suggesting potential problems with the fire emissions during this period.

During winter the model tends to overpredict the observed bbOA values in

Barcelona, Cabauw, Melpitz, Helsinki and Hyytiala. On the other hand, the model
underpredicts the bbOA in Mace Head and Chilbolton by approximately 0.3 μg m$^{-3}$ on
average. The prediction skill metrics of PMCAMx-SR (base case and sensitivity test)
against AMS factor analysis during the modelled periods are also provided in the
Supplementary Information (Tables S2-S5). These problems in reproducing
wintertime OA measurements were also noticed by Denier van der Gon et al. (2015)
and suggest problems in the emissions and/or the simulation of the bbOA during this
cold period with slow photochemistry.

## 8   Conclusions

A source-resolved version of PMCAMx, called PMCAMx-SR was developed and
tested. This new version can be used to study independently specific organic aerosol



sources (eg. diesel emissions) if so desired by the user. We applied PMCAMx-SR to
the European domain during an early summer and a winter period focusing on
biomass burning.
The concentrations of bbOA (sum of bbPOA and bbSOA) and their
contributions to total OA over Europe are, as expected, quite variable in space and
time. During the early summer, the contribution of bbOA to total OA over Europe
was predicted to be 16%, while during winter it increased to 47%. Secondary biomass
burning OA was predicted to be approximately 70% of the bbOA during summer and
only 30% during the winter on average. The production of bbSOA increases the range
of influence of fires.
The IVOCs emitted by the fires can be a major source of SOA. In our
simulations, the IVOCs with saturation concentrations $C^*=10^5$ and $10^6$ μg m$^{-3}$
contributed approximately one third of the average bbOA over Europe. The emissions
of these compounds and their aerosol forming potential are uncertain, so the
formation of bbSOA clearly is an importance topic for future work.
PMCAMx-SR was evaluated against AMS measurements taken at various
European measurement stations and the results of the corresponding PMF analysis.
During the summer the model reproduced without bias the average measured bbPOA
levels in Cabauw and the practically zero levels in Finokalia and Melpitz. However, it
underpredicted the bbPOA in Mace Head. Its performance for oxygenated organic
aerosol (OOA) which should include bbSOA together with a lot of other sources was
mixed: overprediction in Cabauw (fractional bias +42%), Mace Head (fractional bias
+34%), and Finokalia (fractional bias +23%) and underprediction in Melpitz
(fractional bias -14%).
During the winter the model overpredicted the bbPOA levels in most stations
(Cabauw, Helsinki, Melpitz, Hyytiala, Barcelona), while it underpredicted in Mace
Head and Chibolton. At the same time, it reproduced the measured OOA
concentrations with less than 15% bias in Cabauw, Helsinki, and Hyytiala,
underpredicted OOA in Melpitz, Barcelona, and Chibolton and overpredicted OOA in
Mace Head. These results both potential problems with the wintertime emissions of
bbPOA and the production of secondary OA during the winter.



*Data availability*. The data in the study are available from the authors upon request
(spyros@chemeng.upatras.gr).

*Author contributions*. GNT conducted the simulations, analysed the results, and wrote
the paper. SNP was responsible for the design of the study, the synthesis of the results
and contributed to the writing of the paper.
*Competing interests*. The authors declare that they have no conflict of interest.
*Acknowledgements*. This study was financially supported by the European Union's
Horizon 2020 EUROCHAMP–2020 Infrastructure Activity (Grant agreement 730997)
and the Western Regional Air Partnership (WRAP Project No. 178-14).

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



1    **Table 1.** Parameters used to simulate POA and bbPOA emissions in PMCAMx-SR.

| $C^*$ at 298 K ($\mu$g m$^{-3}$) | $10^{-2}$ | $10^{-1}$ | $10^{0}$ | $10^{1}$ | $10^{2}$ | $10^{3}$ | $10^{4}$ | $10^{5}$ | $10^{6}$ |
|---|---|---|---|---|---|---|---|---|---|
| **POA** | | | | | | | | | |
| Fraction of POA emissions[1] | 0.03 | 0.06 | 0.09 | 0.14 | 0.18 | 0.30 | 0.40 | 0.50 | 0.80 |
| Effective Vaporization Enthalpy (kJ mol$^{-1}$) | 112 | 106 | 100 | 94 | 88 | 82 | 76 | 70 | 64 |
| **bbPOA** (Base Case) | | | | | | | | | |
| Fraction of POA emissions | 0.2 | 0.0 | 0.1 | 0.1 | 0.2 | 0.1 | 0.3 | 0.50 | 0.80 |
| Effective Vaporization Enthalpy (kJ mol$^{-1}$) | 93 | 89 | 85 | 81 | 77 | 73 | 69 | 70 | 64 |
| **bbPOA** (Sensitivity Test) | | | | | | | | | |
| Fraction of POA emissions | 0.2 | 0.0 | 0.1 | 0.1 | 0.2 | 0.1 | 0.3 | - | - |
| Effective Vaporization Enthalpy (kJ mol$^{-1}$) | 93 | 89 | 85 | 81 | 77 | 73 | 69 | - | - |

[1]This is the traditional non-volatile POA included in inventories used for regulatory purposes. The sum of all fractions can exceed unity because a large fraction of the IVOCs is not included in these traditional particle emission inventories.



**Table 2.** Organic compound emission rates (in tn d$^{-1}$) over the modeling domain
during the simulated periods.

| | Emission rate (tn d$^{-1}$) |
|---|---|
| **1 – 29 May 2008** | |
| Wildfires | 1,700 |
| Residential | 700 |
| Agriculture - waste burning | 300 |
| **25 February – 22 March 2009** | |
| Wildfires | 3,000 |
| Residential | 6,000 |
| Agriculture - waste burning | 320 |



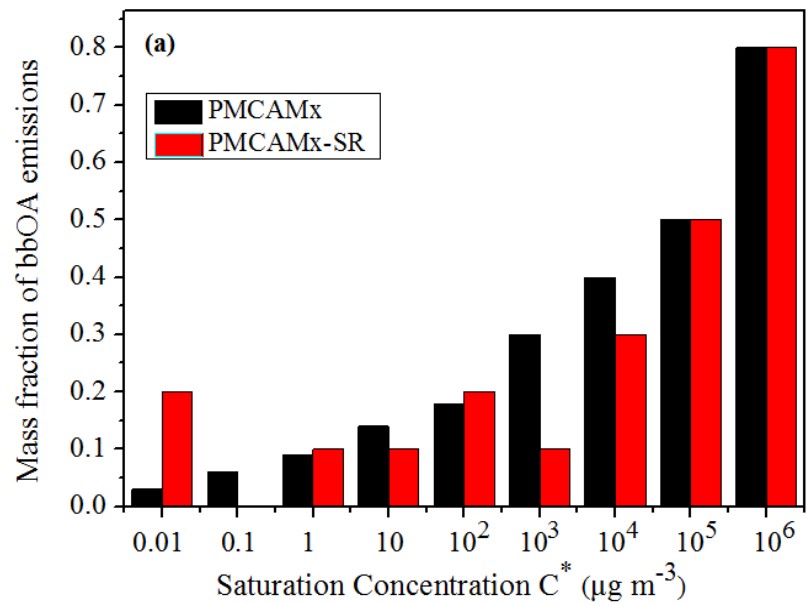

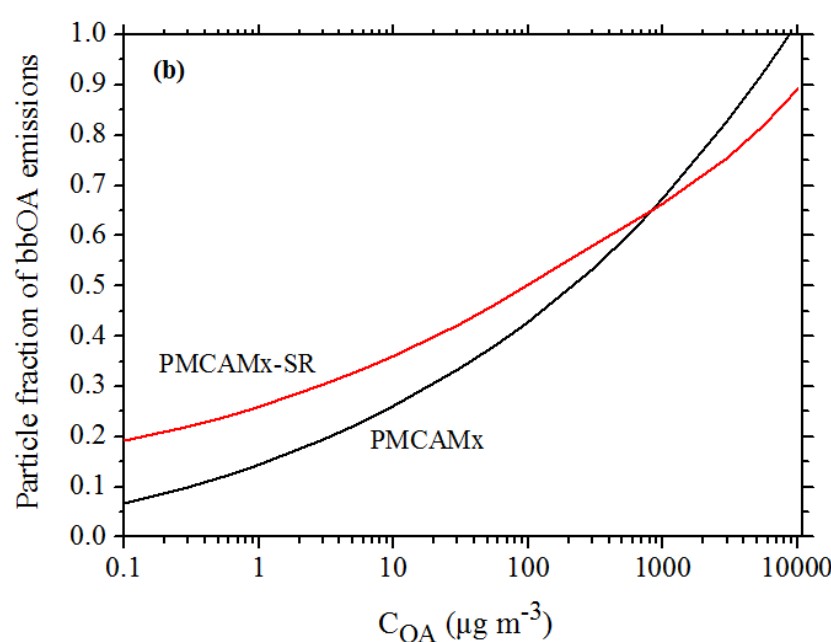

**Figure 1.** (a) Volatility distribution of bbOA in PMCAMx and PMCAMx-SR. (b)
Particle fractions of bbOA emissions as a function of OA concentration at 298 K.





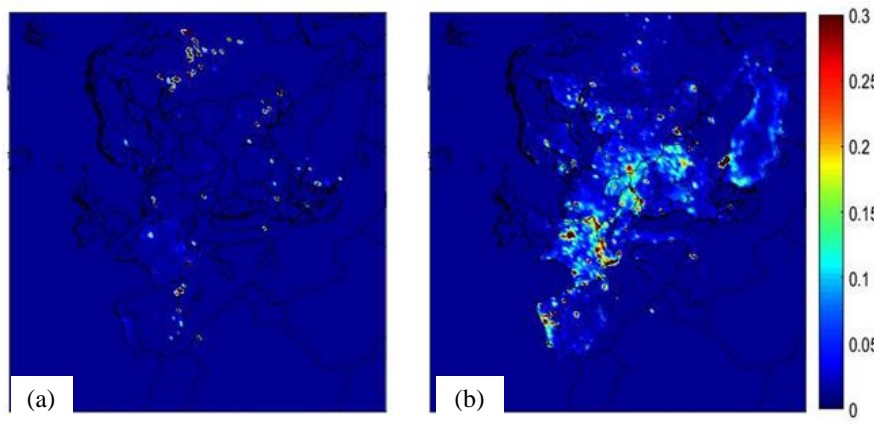

**Figure 2.** Spatial distribution of average biomass burning OA emission rates (kg d$^{-1}$ km$^{-2}$) for the two simulation periods: (a) 1-29 May 2008 and (b) 25 February-22 March 2009.





**Figure 3.** PMCAMx-SR predicted base case ground – level concentrations of PM$_{2.5}$ (a-b) fPOA, (c-d) bbPOA, (e-f) SOA and (g-h) bbSOA, during the modeled summer and winter periods.



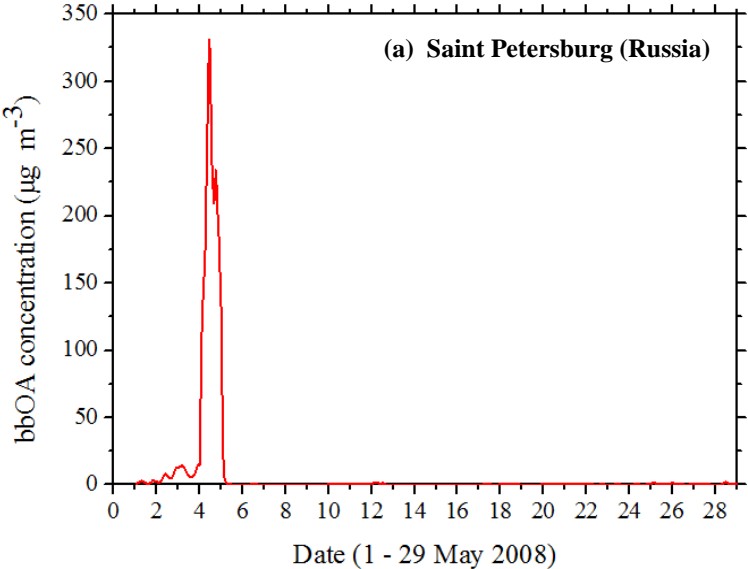

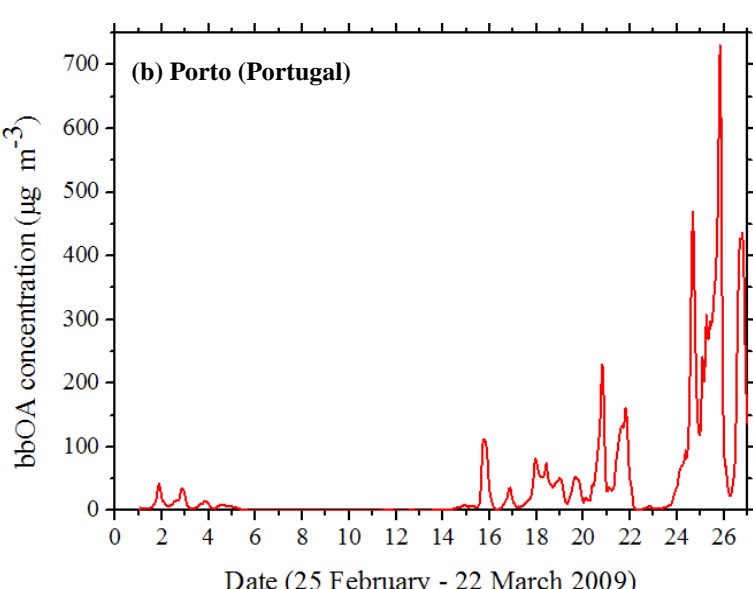

4  **Figure 4.** Timeseries of PM$_{2.5}$ bbOA concentrations in (a) Saint Petersburg in Russia

5  during 1-29 May 2008 and in (b) Porto in Portugal during 25 February-22 March

6  2009.







13 **Figure 5.** PMCAMx-SR predicted base case ground – level concentrations of PM$_{2.5}$

14 bbPOA and bbSOA, during 1 – 6 May 2008 in the Scandinavian Peninsula and

15 Russia.



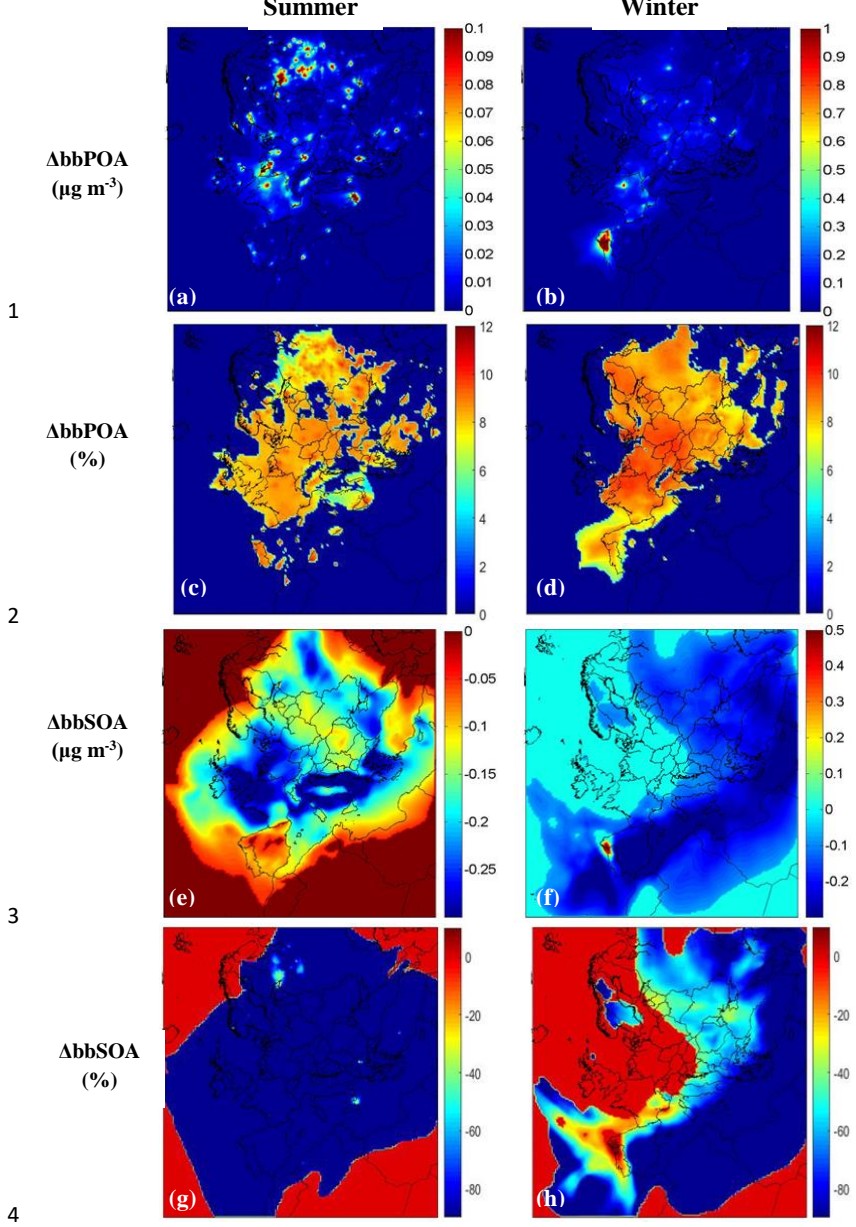

**Figure 6.** Average predicted absolute (µg m$^{-3}$) difference (Sensitivity Case – Base Case) of ground-level PM$_{2.5}$ (a-b) bbPOA and (e-f) bbSOA concentrations from PMCAMx-SR base case and sensitivity simulations during the modeled periods. Also shown the corresponding relative (%) change of ground-level PM$_{2.5}$ (c-d) bbPOA and (g-h) bbSOA concentrations during the modeled periods. Positive values indicate that PMCAMx-SR sensitivity run predicts higher concentrations.



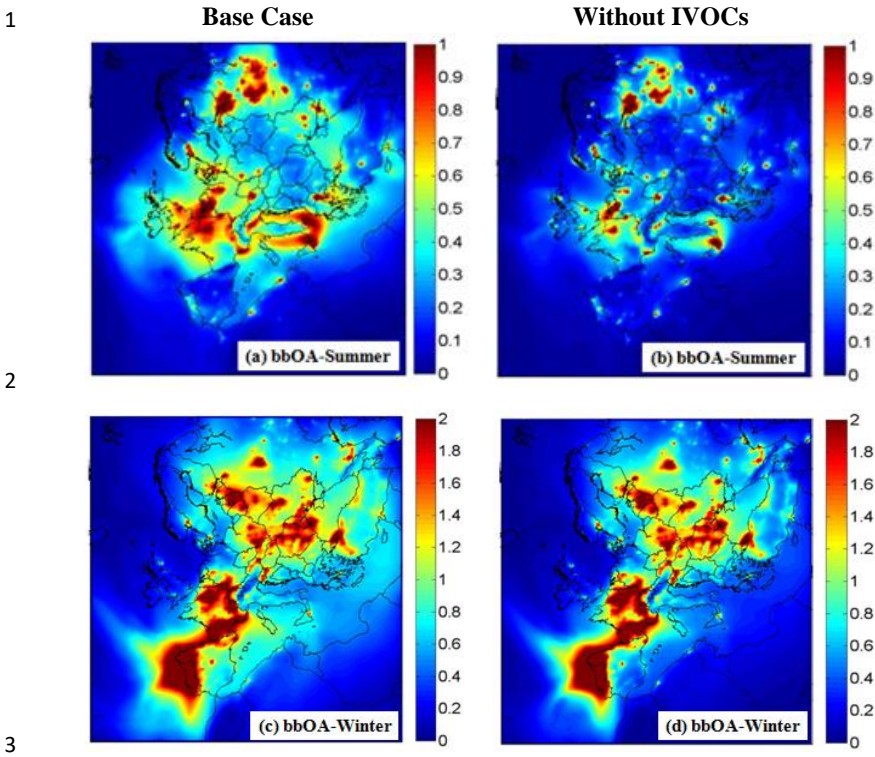

**Figure 7.** Predicted ground-level concentrations of PM$_{2.5}$ total bbOA (μg m$^{-3}$) during the modeled summer (a-b) and the modeled winter (c-d) period. The figures to the left are for the PMCAMx-SR base case simulation while those to the right for the low-IVOC sensitivity test.



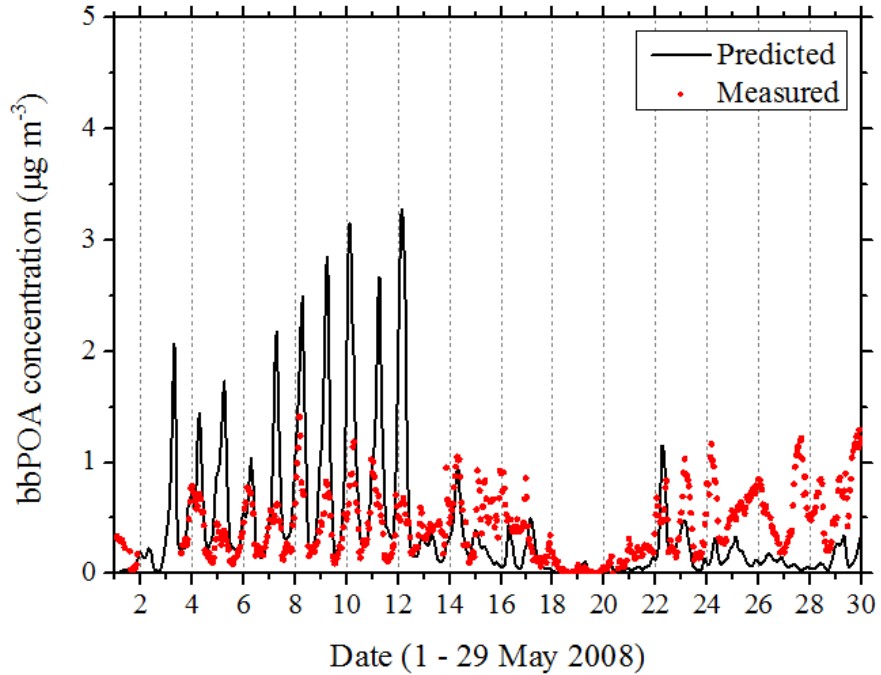

2   **Figure 8.** Comparison of hourly bbPOA concentrations predicted by PMCAMx-SR

3   with values estimated by PMF analysis of the AMS data in Cabauw during 1-29 May

4   2008.