# Peer review of "Simulation of the chemical evolution of biomass burning organic"

_Atmospheric Chemistry and Physics, 2018_

## Referee Comment (RC1) · Anonymous Referee #1 · 11 Jan 2019

This paper describes simulations using the PMCAMx chemical transport model of concentrations of organic aerosol (OA) over Europe for a wintertime period and a summertime period, with the simulated OA concentration fields subdivided according to biomass burning POA, fossil POA, and SOA derived from biomass burning emissions and from other I/S-VOC. The purpose is both to simulate the biomass burning source of OA specifically, but also to be able to apply differential volatility distributions and aging characteristics for the biomass burning source compared with other OA sources. The time periods simulated include instances of very large localised biomass burning emissions, such as wildfires, and winter-time heating emissions in northern Europe.

This is a short and very clearly presented piece of work. Methodological approach and results are clearly described. I do not have any scientific/technical issue with the work. The work adds to the estimations of amount of OA from different origins for the European domain, where that estimation is sometimes derived from 'backward' source-receptor modelling of measurements or, as here, from 'forward' chemical-transport modelling from estimated emissions. Where this paper has some limitation is in 'ground truth-ing' the model simulations. The authors do provide some comparison summary statistics between their model concentrations and those derived from AMS-PMF measurements at a few sites across Europe but it can be difficult to draw conclusions from such comparisons because model and measurement data do not always represent exactly the same chemical/source entity. The authors conclude there is a potential shortcoming in emissions data for residential heating but do not undertake model sensitivities on changing the emissions.

Overall, however, I am happy to recommend this manuscript for publication as it is. I spotted only very few formatting errors:

Line 112: insert "and" before "intermediate".

Line 283: cite to Fig. 4a rather than generically to Fig. 4.

Line 302: after "on March 21" add a citation to Fig. 4b.

---

## Referee Comment (RC2) · Anonymous Referee #2 · 29 Jan 2019

The Theodoritsi and Pandis manuscript reports on the predicted sensitivity of organic aerosol (OA) mass to biomass burning emissions, using a source resolved version of the chemical transport model, PMCAMX (PMCAMX-SR). Studies such as this one are important for understanding the potential air quality and climate effects of anthropogenic and biogenic biomass burning emissions, particularly since the representation of biomass burning-derived SOA is relatively undeveloped in most chemical transport models. The inclusion of biomass-burning derived SOA, particularly when including IVOC, leads to substantial contributions to total predicted OA. This study highlights the need to better constrain biomass burning emissions inventories, including the volatility distribution, and to better understand SOA formation potentials of those emissions.

This manuscript is likely to be of interest to the ACP community, and publication is recommended upon addressing the following comments.

Technical Comments

It is known that simulating the spatial and temporal distribution of OA, particularly SOA, can be challenging; compensating errors can obscure model performance. In the abstract and in section 7, in addition to the absolute performance statistics, it would be useful to report the change in performance with the expanded treatment of bb-OA (POA+SOA). Weaker performance in winter could be a function of the base simulation (emissions, chemistry, and/or meteorology) and not necessarily a function of the expanded treatment of bb.

Line 58: It is recommended that it be emphasized that bbOA is added as a third category, and is not explicitly considered anthropogenic or biogenic, though bb emissions are characterized in the manuscript as anthropogenic (ag. and heating) or biogenic (wildfire).

Line 111-113: How many model compounds are used to represent IVOCs and SVOCs, respectively? Was the SAPRC mechanism updated as part of this study? If so, the authors should provide further detail in the supplement. If not, a reference should be provided (may be the Environ reference, just needs to be moved).

Line 139: Was the May et al. volatility distribution applied to all bb emissions? The use of "wood burning" here implies only residential wood burning, but it is assumed that the bb volatility distribution was applied to all three categories of bb emissions. This needs to be modified/clarified.

Line 146: How does partitioning within this model framework depend on aerosol composition?

Lines 153-155: The description of the Lane et al. VBS scheme is confusing as written. Given the generally widespread use of the VBS SOA model, it might be clearer to

write that SOA is represented using 4 bins, and X number of VOC precursors that are tracked separately as either aSOA-v or bSOA-v. So the number of actual model surrogates seems like it would only be 4*aSOA-v,gas + 4*aSOA-v,p +4*bSOA-v,gas + 4*bSOA-v,p, and is not dependent of the number of VOC precursors (as implied by 4 surrogate SOA compounds per VOC).

Lines 162-170: The description of chemical aging is also somewhat confusing. It might be clearer to refer to the volatility bin, rather than "vapors" and "semi-volatile SOA". Do the POA and SOA aging reactions both result in an increase in OA mass (line 170)? Is this independent of the mass increase associated with a shift to a semi-volatile bin? Does the OA mass increase apply to the biogenic SOA aging, even though no change in volatility is assumed?

To clarify the volatility distributions and aging, a figure such as 5-2 in the CAMx user's guide would be very helpful.

Section 2.2: It is recommended that the authors consider the publication by Alvarado et al. (2015), which also evaluated volatility distributions of bb emissions. It may be beyond the scope of the manuscript to repeat the model runs using the Alvarado volatility distribution, but it would be useful to consider it in the introduction and discussion, and include it in the Figure 1 panels. The Alvarado et al. study also attempted to account for IVOC emissions not included in two published volatility distributions (including May et al.). Overall, there is significantly more mass (or higher fraction of bb-POA emissions) in the 105 and 106 bins in the subject manuscript (base case) than in Alvarado et al.

Also, while scaling the anthropogenic POA EF by 1.5, which gives a sum of fractions >1, has been well described in current literature, it is not clear that the same rationale applies to the biomass burning emissions used in this work. While the IVOC bins are not constrained by data and thus absent in the published VBS distributions, this is not equivalent to missing mass in the bb-POA emissions totals. It seems that some scaling of the May et al. fractions may be needed to include the IVOC bins without giving a

sum of fractions >1 (e.g., as done in Alvarado et al.). This probably needs a bit more discussion/clarification in the methods, as the mass attributed to the IVOC bins has a significant effect on predictions of bb-SOA (as demonstrated by the sensitivity case). Reference: Alvarado et al., ACP, 15: 6667-6668, doi:10.5194/acp-15-6667-2015

Editorial Comments

In general, it is recommended that the authors check carefully for use of abbreviations. In many instances, an abbreviation is introduced but then not used consistently throughout the manuscript (e.g., organic aerosol (OA) in section 2.1). In a few cases, an abbreviation is introduced but not defined (e.g., AMS line 100).

Line 16: Oxidation products of the bbOA? Or of bb emissions? If the latter, sentence needs revision.

Line 22: Suggest removing "same" before contribution. It is a little confusing as written.

Line 50: What does "their" refer to?

Lines 54 and 57: Suggest using "or" rather than "and", to indicate OA can be primary *or* secondary and of anthropogenic *or* biogenic origin.

Lines 217-232: The discussion about the emissions is a bit unorganized. Are the anthropogenic biomass burning emissions from a source other than GEMS or the Pan-European inventory? If not, recommend to add "including anthropogenic biomass burning emissions" (line 203 or 210). Line 218-219 is then not needed. It is also recommended to move line 217 to the previous paragraph in which the other emissions inventories are described (likely before the introduction of MEGAN).

Italicize variables in equations.

---

## Author Response (AR1)

**Responses to the Comments of Referee 1**

(1) This paper describes simulations using the PMCAMx chemical transport model of concentrations of organic aerosol (OA) over Europe for a wintertime period and a summertime period, with the simulated OA concentration fields subdivided according to biomass burning POA, fossil POA, and SOA derived from biomass burning emissions and from other I/S-VOC. The purpose is both to simulate the biomass burning source of OA specifically, but also to be able to apply differential volatility distributions and aging characteristics for the biomass burning source compared with other OA sources. The time periods simulated include instances of very large localised biomass burning emissions, such as wildfires, and winter-time heating emissions in northern Europe.

This is a short and very clearly presented piece of work. Methodological approach and results are clearly described. I do not have any scientific/technical issue with the work. The work adds to the estimations of amount of OA from different origins for the European domain, where that estimation is sometimes derived from 'backward' source-receptor modelling of measurements or, as here, from 'forward' chemical-transport modelling from estimated emissions. Where this paper has some limitation is in 'ground truthing' the model simulations. The authors do provide some comparison summary statistics between their model concentrations and those derived from AMS-PMF measurements at a few sites across Europe but it can be difficult to draw conclusions from such comparisons because model and measurement data do not always represent exactly the same chemical/source entity. The authors conclude there is a potential shortcoming in emissions data for residential heating but do not undertake model sensitivities on changing the emissions.

Overall, however, I am happy to recommend this manuscript for publication as it is. I spotted only very few formatting errors:

We appreciate the positive feedback from the referee. Indeed, the evaluation of the ability of PMCAMx-SR to reproduce the biomass burning OA is necessarily limited, because of the lack of the corresponding necessary measurements. Comparisons of total OA measurements and model predictions are difficult to interpret, because there are so many OA sources. We did our best using the available estimated biomass burning OA concentrations from the analysis of the Aerosol Mass Spectrometer measurements during the period (see for example Figure 8). Improving the OA emission estimates from residential heating in Europe is a major undertaking and it is clearly beyond the scope of the present work. Such an effort is described by Denier van der Gon et al. (ACP, 15, 6503-6519, 2015). However, we clearly need to do better and this requires a good pan-European OA measurement dataset that is currently lacking. We have added some discussion about this important model evaluation issue in the revised paper.

(2) Line 112: insert "and" before "intermediate". Done.

(3) Line 283: cite to Fig. 4a rather than generically to Fig. 4. Corrected.

(4) Line 302: after "on March 21" add a citation to Fig. 4b. Done.

**Responses to the Comments of Referee 2**

(1) The Theodoritsi and Pandis manuscript reports on the predicted sensitivity of organic aerosol (OA) mass to biomass burning emissions, using a source resolved version of the chemical transport model, PMCAMX (PMCAMX-SR). Studies such as this one are important for understanding the potential air quality and climate effects of anthropogenic and biogenic biomass burning emissions, particularly since the representation of biomass burning-derived SOA is relatively undeveloped in most chemical transport models. The inclusion of biomass-burning derived SOA, particularly when including IVOC, leads to substantial contributions to total predicted OA. This study highlights the need to better constrain biomass burning emissions inventories, including the volatility distribution, and to better understand SOA formation potentials of those emissions. This manuscript is likely to be of interest to the ACP community, and publication is recommended upon addressing the following comments.

We do appreciate the constructive comments and suggestions of the referee. We have done our best to address them and revise the paper accordingly. Our responses (regular font) follow the comments of the referee (in italics) below.

**Technical Comments:**

(2) It is known that simulating the spatial and temporal distribution of OA, particularly SOA, can be challenging; compensating errors can obscure model performance. In the abstract and in section 7, in addition to the absolute performance statistics, it would be useful to report the change in performance with the expanded treatment of bb-OA (POA+SOA). Weaker performance in winter could be a function of the base simulation (emissions, chemistry, and/or meteorology) and not necessarily a function of the expanded treatment of bbOA.

We agree with the referee, that the performance of a model for total OA depends on a lot of factors (multiple sources, secondary production, removal, meteorology). The PMCAMx-SR performance is a little better for OA than that of the regular PMCAMx. PMCAMx has a similar weak performance in the winter suggesting, as the reviewer states, that this is probably due to other factors and not the expanded treatment of the bbOA. We have added a new section discussing briefly the differences in performance and also the above issues.

(3) Line 58: It is recommended that it be emphasized that bbOA is added as a third category, and is not explicitly considered anthropogenic or biogenic, though bb emissions are characterized in the manuscript as anthropogenic (ag. and heating) or biogenic (wildfire).

We have clarified this point, following the reviewer's suggestion. Indeed, the model does not currently separate the anthropogenic (e.g., residential heating, agricultural burning) from the biogenic sources (wildfires) of bbOA.

(4) Line 111-113: How many model compounds are used to represent IVOCs and SVOCs, respectively? Was the SAPRC mechanism updated as part of this study? If so, the authors should provide further detail in the supplement. If not, a reference should be provided (may be the Environ reference, just needs to be moved).

In this work the IVOCs, SVOCs, and LVOCs are described with 9 volatility bins  $(10^{-2} - 10^6 \,\mu g \,m^{-3})$ . Different lumped compounds are used for the fresh (primary) and secondary organic compounds. The simple reactions of these compounds (one volatility bin change for each reaction with OH) have been added the original SAPRC mechanism. The size distributions of the SVOCs and LVOCs in the particulate phase are described. This information has been added to the revised paper.

(5) Line 139: Was the May et al. volatility distribution applied to all bb emissions? The use of "wood burning" here implies only residential wood burning, but it is assumed that the bb volatility distribution was applied to all three categories of bb emissions. This needs to be modified/clarified.

The May et al. volatility distribution was applied to all three categories of bb emissions. The use of the term "wood burning" in line 139 is now replaced with "biomass burning". We also clarify that the same volatility distribution is assumed for all bbOA sources.

**(6) Line 146: How does partitioning within this model framework depend on aerosol composition?**

Organic gas-particle partitioning depends on aerosol composition according to gasparticle partitioning absorption theory. The model assumes that the organic compounds form a single pseudo-ideal solution in the particle phase and do not interact with the aqueous phase. This is now clarified in the paper and a reference is provided.

(7) Lines 153-155: The description of the Lane et al. VBS scheme is confusing as written. Given the generally widespread use of the VBS SOA model, it might be clearer to write that SOA is represented using 4 bins, and X number of VOC precursors that are tracked separately as either aSOA-v or bSOA-v. So the number of actual model surrogates seems like it would only be 4\*aSOA-v,gas + 4\*aSOA-v,p +4\*bSOA-v,gas +4\*bSOA-v,p, and is not dependent of the number of VOC precursors (as implied by 4 surrogate SOA compounds per VOC).

We have followed the reviewer's suggestion and provided additional information to avoid confusing. Based on the original work of Lane et al. (2008a), SOA from VOCs is represented using four volatility bins (1, 10,  $10^2$ ,  $10^3 \ \mu g \ m^{-3}$  at 298 K). As the reviewer suggests the model uses 4 surrogate compounds for SOA from anthropogenic VOCs (aSOA-v) and another 4 for SOA from biogenic VOCs (bSOA-v). These can exist in either the gas or particulate phase so there are two variables from each. There additional surrogate compounds for the oxidation products of anthropogenic IVOCs and SVOCs. PMCAMx-SR includes additional SOA surrogate compounds from biomass burning. We have followed the suggestion of the reviewer and clarified the SOA VBS-scheme in the revised manuscript.

(8) Lines 162-170: The description of chemical aging is also somewhat confusing. It might be clearer to refer to the volatility bin, rather than "vapors" and "semi-volatile SOA". Do the POA and SOA aging reactions both result in an increase in OA mass (line 170)? Is this independent of the mass increase associated with a shift to a semi-volatile bin? Does the OA mass increase apply to the biogenic SOA aging, even though no change in volatility is assumed?

We have rephrased these sentences to clarify that the aging reactions are for the material of each volatility bin that is in the gas phase. We clarify that all these aging reactions (both POA and SOA) are assumed to reduce the volatility of the reacted vapour by one order of magnitude which is linked to an increase in OA mass by approximately 8 percent to account for added oxygen. For the biogenic SOA aging is assumed to lead to no net change of volatility and OA mass. This is also clarified now.

(9) To clarify the volatility distributions and aging, a figure such as 5-2 in the CAMx user's guide would be very helpful.

This is a good idea. We have added a new figure to the paper (Figure 2) depicting the various OA components simulated and their chemical aging reactions.

(10) Section 2.2: It is recommended that the authors consider the publication by Alvarado et al. (2015), which also evaluated volatility distributions of bb emissions. It may be beyond the scope of the manuscript to repeat the model runs using the Alvarado volatility distribution, but it would be useful to consider it in the introduction and discussion, and include it in the Figure 1 panels. The Alvarado et al. study also attempted to account for IVOC emissions not included in two published volatility distributions (including May et al.). Overall, there is significantly more mass (or higher fraction of bb-POA emissions) in the  $10^5$  and  $10^6$  bins in the subject manuscript (base case) than in Alvarado et al.

We have followed the recommendation of the reviewer and added some discussion of the Alvarado et al. volatility distribution. We also added the results of the Alvarado study in the discussion of the role of IVOCs from biomass burning in SOA formation.

(11) Also, while scaling the anthropogenic POA EF by 1.5, which gives a sum of fractions >1, has been well described in current literature, it is not clear that the same rationale applies to the biomass burning emissions used in this work. While the IVOC bins are not constrained by data and thus absent in the published VBS distributions, this is not equivalent to missing mass in the bb-POA emissions totals. It seems that some scaling of the May et al. fractions may be needed to include the IVOC bins without giving a sum of fractions >1 (e.g., as done in Alvarado et al.). This probably needs a bit more discussion/clarification in the methods, as the mass attributed to the IVOC bins has a significant effect on predictions of bb-SOA (as demonstrated by the sensitivity case). Reference: Alvarado et al., ACP, 15: 6667-6668, doi:10.5194/acp-15-6667-2015

This is often a confusing point. We have added discussion clarifying it. We do underline the difference with the Alvarado et al. emissions in the corresponding discussion. We also include the absolute emission rates for all bins in Table S1 to make sure that there is no confusion about the used input in our simulation.

**Editorial Comments:**

(12) In general, it is recommended that the authors check carefully for use of abbreviations. In many instances, an abbreviation is introduced but then not used consistently throughout the manuscript (e.g., organic aerosol (OA) in section 2.1). In a few cases, an abbreviation is introduced but not defined (e.g., AMS line 100).

We have reviewed all the abbreviations used in the manuscript and made the corresponding corrections.

(13) Line 16: Oxidation products of the bbOA? Or of bb emissions? If the latter, sentence needs revision.

We have revised this sentence. These are the oxidation products of biomass burning emissions.

(14) Line 22: Suggest removing "same" before contribution. It is a little confusing as written.

Corrected.

(15) Line 50: What does "their" refer to?

It refers to the organic compounds. We have rephrased this sentence.

(16) Lines 54 and 57: Suggest using "or" rather than "and", to indicate OA can be primary or secondary and of anthropogenic or biogenic origin. Done.

(17) Lines 217-232: The discussion about the emissions is a bit unorganized. Are the anthropogenic biomass burning emissions from a source other than GEMS or the Pan-European inventory? If not, recommend to add "including anthropogenic biomass burning emissions" (line 203 or 210). Line 218-219 is then not needed. It is also recommended to move line 217 to the previous paragraph in which the other emissions inventories are described (likely before the introduction of MEGAN). We have made the corresponding corrections to organize better the discussion about

the emissions used in this study.

(18) Italicize variables in equations. Done.

[revised manuscript text omitted]

201 evaporation of POA are assumed to react with OH radicals in the gas phase with a rate constant of  $k = 4 \times 10^{-11}$  cm3 molec-1 s-1 resulting in the formation of oxidized 202 203 OAlower volatility aSOA. These reactions are assumed to lead to an effective reduction of volatility by one order of magnitude. Semi-volatile anthropogenic aSOA-204 205 v components are also assumed to react with OH in the gas phase with a rate constant of  $k = 1 \times 10^{-11}$  cm3 molec-1 s-1 for anthropogenic SOA (Atkinson and Arev, 2003). 206 All these aging reactions (both POA and SOA) are assumed to reduce the volatility of 207 the reacted vapour by one order of magnitude, which is linked to an increase in OA 208 209 mass by approximately 7.5% to account for added oxygen. Biogenic SOA (bSOA-v) 210 aging is assumed to lead to zero net change of volatility and OA mass (Lane et al., 211 2008b). Each reaction is assumed to increase the OA mass by 7.5% to account for 212 added oxygen.

213

**214 2.2 PMCAMx-SR enhancements**

215 In PMCAMx-SR, the fresh biomass burning organic aerosol (bbOA) and its secondary oxidation products (bbSOA) are simulated separately from the other POA 216 components. The May et al. (2013) volatility distribution is used to simulate the gas-217 particle partitioning of fresh bbOA. This distribution includes surrogate compounds 218 up to a volatility of  $10^4 \,\mu g \, m^{-3}$ . This means that the more volatile IVOCs, which could 219 contribute to SOA formation, are not included. To close this gap, the values of the 220 volatility distribution of Robinson et al. (2007) are used for the  $10^5$  and  $10^6 \,\mu g m^{-3}$ 221 bins (Table 1). The sensitivity of PMCAMx-SR to the IVOC emissions added to the 222 223 May et al. (2013) distribution will be explored in a subsequent section. The effective 224 saturation concentrations and the enthalpies of vaporization used for bbOA in PMCAMx-SR are also listed in Table 1. The new bbOA scheme requires the 225 introduction of 36 new organic species to simulate both phases of fresh primary and 226 oxidized bbOA components. The rate constant used for the chemical aging reactions 227 is the same as the one currently used for all POA components and has a value of k = 4228  $\times$  10-11 cm3 molec-1 s-1. The volatility distributions of bbOA in PMCAMx and 229 PMCAMx-SR are shown in Fig. 1a. The volatility distribution implemented in 230 PMCAMx-SR results in less volatile bbOA for ambient OA levels (a few µg m-3) 231 232 (Fig. 1b). Figure 2 is a A schematic representation of the proposed organic aerosol frameworkmodule implemented in of -PMCAMx and -PMCAMx-SR is shown in 233

234 Figure 2. depicting the various OA components simulated and their chemical aging
 235 reactions.

[revised manuscript text omitted]

393 Alvarado et al. (2015) also evaluated the volatility distribution of bb emissions by using the Aerosol Simulation Program (ASP) to simulate the chemical evolution of 394 SOA within a young biomass burning smoke plume sampled over California in 395 November 2009. SOA formation is simulated using the VBS scheme as proposed by 396 Robinson et al. (2007) and the gas-phase chemistry implemented for organic 397 compounds was RACM2. The volatility distribution for the POA was taken from the 398 399 wood smoke study of Grieshop et al. (2009a) and implies that most of the total mass of the organic compounds species is in the aerosol phase which leads to OA 400 overestimation. However, Grieshop et al. (2009a) was only able to measure species 401

402 with a saturation mass concentration of  $10^4 \ \mu g \ m^3$  or less. To account for the 403 unidentified IVOCs an additional organic mass was included in the C\*=105  $\mu g \ m^3$  and 404 C\*=106  $\mu g \ m^3$  which lead to a reduction of the OA formed. Alvadaro et al. (2015) 405 concluded that the unidentified IVOCs are mainly more volatile (C\*=106  $\mu g \ m^3$ ). 406

[revised manuscript text omitted]